# *CAD* Genes: Genome-Wide Identification, Evolution, and Their Contribution to Lignin Biosynthesis in Pear (*Pyrus bretschneideri*)

**DOI:** 10.3390/plants10071444

**Published:** 2021-07-15

**Authors:** Kaijie Qi, Xiaofei Song, Yazhou Yuan, Jianping Bao, Xin Gong, Xiaosan Huang, Shahrokh Khanizadeh, Shaoling Zhang, Shutian Tao

**Affiliations:** 1Pear Engineering Research Centre, College of Horticulture, Nanjing Agricultural University, 1 Weigang, Nanjing 210095, China; qikaijie@njau.edu.cn (K.Q.); 2015104031@njau.edu.cn (X.S.); 2016104031@njau.edu.cn (Y.Y.); 2016104030@njau.edu.cn (X.G.); huangxs@njau.edu.cn (X.H.); zhangsl@njau.edu.cn (S.Z.); 2College of Plant Science, Tarim University, Ala’er City 843300, China; baobao-xinjiang@126.com; 3ELM Consulting Inc., St-Lazare, QC J7T, Canada; Shahrokh.Khanizadeh@outlook.com

**Keywords:** cinnamyl alcohol dehydrogenase, gene family, Rosaceae, stone cell, expression

## Abstract

The synthetic enzyme cinnamyl alcohol dehydrogenase (CAD) is involved in responses to various stresses during plant growth. It regulates the monolignol biosynthesis and catalyzes hydroxyl cinnamaldehyde reduction to the corresponding alcohols. Although the CAD gene families have been explored in some species, little known is in Rosaceae. In this study, we identified 149 genes in *Pyrus bretschneideri* (*PbrCAD*), *Malus domestica* (*MDPCAD*), *Prunus mume* (*PmCAD*) and *Fragaria vesca* (*mrnaCAD*). They were phylogenetically clustered into six subgroups. All CAD genes contained ADH-N and ADH-zinc-N domains and were distributed on chromosomes unevenly. Dispersed and WGD/segmental duplications accounted the highest number of evolutionary events. Eight collinear gene pairs were identified among the four Rosaceae species, and the highest number was recorded in pear as five pairs. The five *PbrCAD* gene pairs had undergone purifying selection under Ka/Ks analysis. Furthermore, nine genes were identified based on transcriptomic and stone cell content in pear fruit. In qRT-PCR, the expression patterns of *PbrCAD1*, *PbrCAD20*, *PbrCAD27*, and *PbrCAD31* were consistent with variation in stone cell content during pear fruit development. These results will provide valuable information for understanding the relationship between gene expressions and stone cell number in fruit.

## 1. Introduction

In plants, the phenolic polymer lignin plays crucial roles in water retention, mechanical support, and protection [1,2,3]. However, the content of lignin also greatly affects the quality of some fruits, reducing the flavor and quality [4,5]. The process of cell wall lignification involves in many enzymes and corresponding genes [6,7]. Cinnamyl alcohol dehydrogenase (CAD) is a key enzyme in the lignin biosynthesis, and the activity of CAD affects the lignin content and monomer composition [8,9]. CAD catalyzes the reduction of p-coumaraldehyde, coniferaldehyde, and sinapaldehyde to the corresponding hydroxycinnamyl alcohols (monolignols) under the NADPH, then further transformed into lignin as p-hydroxyphenyl (H), guaiacyl (G), and syringyl (S) units [10,11,12,13]. Until now, the *CAD* gene families and their homologs have been identified in various plants [12,14,15,16,17,18].

Pear (*Pyrus*) is the third largest species of fruit tree and is very popular among consumers. Pear fruits have high nutritional value and many medicinal properties [19]. The presence of sclereids is an important factor that affects the pear quality [20]. The deposition of lignin in the cell wall forms the stone cells in pear, and the lignin content in stone cells is over 29.8% [5]. More and more genes in the lignin synthesis have been identified in pears after the completion of pear genome sequencing and assembly, such as cinnamate 4-hydroxylase base pai (C4H), hydroxycinnamoyl-coenzyme A shikimate/quinate hydroxycinnamoyl transferase (HCT), cinnamoyl-CoA reductase (CCR), etc. Furthermore, no studies have identified the *CAD* gene family in lignin biosynthesis in pear.

In this study, we identified the members of the *CAD* gene family in the pear and three other Rosaceae species. We also analyzed the phylogenetic relationships, structures, conserved motifs, synteny, ontology and expression patterns of these genes with the transcriptomic data of *PbrCAD* gene family.

## 2. Results

### 2.1. Identification of the CAD Gene Family in *Rosaceae*

In this study, a total of 234 candidate *CAD* genes were identified. However, 85 candidate genes without CAD domains were discarded using the Pfam and SMART databases. Consequently, 149 CAD genes were identified in Rosaceae, including 57 in pear, 34 in Chinese plum, 42 in apple, and 16 in strawberry, named as *PbrCAD1*-*PbrCAD57*, *MDPCAD1*-*MDPCAD42*, *PmCAD1*-*PmCAD34*, and mrna*CAD1*-*mrnaCAD16*, respectively (Table 1 and Appendix A).

### 2.2. Phylogenetic Analysis of the CAD Genes

The CAD proteins were classified into six main families (Figure 1), designated as clade 1 to 6: C1 (35 proteins), C2 (30), C3 (27), C4 (25), C5 (18), and C6 (14). Furthermore, we detected the specific domains in different colors to mark the branches of genes with particular domains (Figure 1). Each subgroup harbored at least one specific domain without C5. The C1 subgroup also harbored four specific domains and the percentage of members in this group was higher than other subgroups. Three genes in C3 contained the same specific domain (adh short) and they were adjacent in the phylogenetic tree. The remaining two subgroups only contained one specific domain (Figure 1).

### 2.3. Expansion and Evolution of the CAD Gene Family

Each member was assigned to one of the five different categories, as WGD/segmental, singleton, tandem, proximal, or dispersed. In this study, three types of duplication events contributed to the expansion of *CAD* gene family (Table 2 and Appendix A). There were three duplication modes in the pear and only two duplication types in other three Rosaceae species. The dispersed *CAD* genes in apple (95.2%), Chinese plum (58.8%), pear (43.9%), and strawberry (43.8%) accounted for more than half of the total number of genes. However, 56.2% (9) *CAD* genes in strawberry and 42.1% (24) in pear were duplicated and retained from WGD/segmental duplication compared with only 38.2% (13) and 4.8% (two) in apple. The higher proportion of WGD/segmental duplication in pear and strawberry might be attributed to the recent lineage-specific WGD events (30–45 MYA) [21]. Thus, dispersed and WGD/segmental duplications played critical roles in the expansion of *CAD* gene family.

### 2.4. Chromosome Distribution and Synteny Events

Here, the *PbrCAD* genes were distributed randomly on the 11 of 17 pear chromosomes (Figure 2 and Appendix A). A total of 53 *PbrCAD* genes were mapped on chromosomes and four on scaffolds. Chromosome 10 contained the highest number of *PbrCAD* genes (10 genes), whereas chromosomes 3 and 14 exhibited only one. Similar to the *PbrCAD* genes, the distributions of the *CAD* genes in other three Rosaceae genomes were non-uniform (Figure 2 and Appendix A).

In this study, we segmentally searched the duplicated blocks within the genomes of four Rosaceae species. There were eight collinear gene pairs in the four Rosaceae species, including five pairs in pear (*PbrCAD19*-*PbrCAD22*, *PbrCAD24*-*PbrCAD35*, *PbrCAD25*-*PbrCAD27*, *PbrCAD26*-*PbrCAD29*, and *PbrCAD39*-*PbrCAD50*), three pairs in apple (*MDPCAD3-MDPCAD20*, *MDPCAD10*-*MDPCAD23*, and *MDPCAD11*-*MDPCAD24*), and one pair in Chinese plum (*PmCAD15*-*PmCAD29*). However, the *MDPCAD3* was located on an unanchored scaffold (Figure 2).

Furthermore, we calculated the Ka/Ks ratio for the five *PbrCAD* gene pairs to assess the selection pressure among the duplicated *PbrCAD* genes. All the Ka/Ks ratios were <1 (Table 3), implying that they had undergone strong purifying selection and played a critical role in evolution of the *CAD* genes [22].

### 2.5. CAD Gene Structure and CAD Protein Motif Analyses in Pear

In this study, the *PbrCAD* genes were divided into seven subgroups (A1–A7) with high bootstrap support. The 57 *PbrCAD* genes were unequally distributed in each subgroup (Figure 3a). For instance, the A5 subgroup was the largest with 13 members, whereas the A6 subgroup only included four members (Figure 3a).

An exon-intron analysis was performed on the *PbrCAD* genes to gain insight into the diversity of the structure of these genes (Figure 3b and Appendix A). All of the *PbrCAD* genes contained both exons and introns. Moreover, the majority members in the same subgroup contained the similar exon-intron organizations. Most *PbrCAD* genes in subgroup A2 contained five exons and four introns, while those in subgroup A3 contained six exons and five introns. The *PoptrCAD* gene family exhibited three different gene structures [14]. These three patterns (I, II, III) were composed of 5, 5, and 6 exons, respectively, and patterns I and II differred in the length of exons 3 and 4, with exon 3 being longer than exon 4 belongs in pattern I. Patterns I and II were generally present in eudicots and monocots, while pattern III was found in eudicots and magnolia plants, as well as in gymnosperms [14]. We discovered that eight genes belonged to pattern I, 10 genes belonged to pattern II, and 13 genes belonged to pattern III in the *PbrCAD* gene family (Figure 3b and Appendix A), accounting for more than 50% of all genes.

Subsequently, a total of 25 conserved motifs were identified, and the logos of these motifs were showed in Appendix A. Most of the closely related members within the same subgroup had a similar motif composition and arrangement (Figure 3c). Among these motifs, motifs 1, 6, and 8 were detected in most of the PbrCAD genes, indicating that these motifs were the major conserved domains of the PbrCAD family.

### 2.6. Histochemical Test and the Content of Stone Cells during Fruit Development

The pear contained a low stone cell content at the early stage of fruit development (Figure 4). Subsequently, it reached a peak at the middle stage of development, and finally declined gradually over the course of fruit maturation. The stone cell clusters were mainly distributed in the fruit core and pericarp through the staining of longitudinal and transverse sections of fruit revealed (Figure 5). Similar to the trend of stone cell contents, the distribution of stone cell clusters reached the maximum at the middle stage of development. It can be inferred that the middle stage of fruit development is a key period for the formation of lignin and stone cell.

### 2.7. Expression Patterns of the PbrCAD Genes

In this study, 51 *PbrCAD* genes were expressed in pear fruit through the development transcriptome sequencing (RNA-Seq) data (Appendix A) [19,20]. Nine genes exhibited high and stage-specific expressions during the development of the pear fruit (Figure 6 and Appendix A). Subsequently, four of the seven genes were consistent with the RNA-Seq data (without *PbrCAD39*, *PbrCAD41*, and *PbrCAD43*) by the qRT-PCR analysis (Figure 7).

*PbrCAD1*, *PbrCAD20*, *PbrCAD27*, and *PbrCAD31* were expressed at lower levels during the early stage of fruit development, achieved the highest levels of expression at the middle stage, and decreased gradually at the fruit-ripening stage. Furthermore, compared with the physiological data (Figure 5 and Figure 6), the relative expressions of the four genes were in agreement with the changes in lignin physiology. These results demonstrated that these four genes may be involved in the regulation of lignin synthesis in pear fruit.

## 3. Discussion

In our study, the distribution of the *CAD* genes identified from Rosaceae was unequal. A total of 57 *PbrCAD* genes, 42 *MDPCAD* genes, 36 *PmCAD* genes, and 16 *mrnaCAD* genes were discovered. The *CAD* gene family in pear was larger than that of the three other Rosaceae species, which may be related to the peculiar traits of the presence of stone cells in pear.

The phylogenetic tree classified the CAD proteins into six subgroups: C1–C6. Interestingly, each subgroup involved at least one specific domain, with the exception of clade 5. The different modes of gene duplications were important for genomic rearrangement and expansion, and the diversification of gene function [23]. The dispersed duplication and WGD/segmental duplication played critical roles in the expansion of the *CAD* gene family. The insertion and mediation of transposons on distant single-gene translocations may explain the wide spread of the dispersed duplicates [24], which accounted for 43.9% (25) of the *PbrCAD* genes. Our chromosomal-location analysis showed that the distribution of the *CAD* genes in Rosaceae genomes had different densities. The Ka/Ks ratios of five collinear *CAD* gene pairs were <1, suggesting that these genes were evolved under the influence of purifying selection.

We explored the phylogenetic tree, gene structure, and conserved motifs to identify the evolutionary relationship of the *CAD* gene family in pear. These *PbrCAD* genes were classified into seven subgroups, most of the adjacent members contained similar exon-intron organizations and motifs in the same subgroups. *Populus* exhibits three different *CAD* gene structure patterns [14], and *PbrCAD* gene structures accounted for more than 50%. Otherwise, the *PbrCAD* genes contained an additional seven gene structures. In addition, *Oryza* had fewer *CAD* genes than *Populus* and exhibited the greatest number of intron-exon structure variants, which may be attributed to the insertion of transposable elements [17]. This was demonstrated by the dating of the great number of dispersed duplication events in the *PbrCAD* genes, as the transposons increased the dispersed duplication events by regulating distant single-gene translocations [24].

The content of stone cells reached a maximum value and the distribution of the stone cell mass also reached a maximum density at the middle development stage. According to previous research, more than 75% of the total lignin content in fruit pulp is located in mature stone cells [5]. In our study, the stone cell content exhibited a rise-fall tendency, which was consistent with the results reported by Tao et al. (2015) and Ma et al. (2017) [22,25]. The inhibition of the synthesis of lignin in pear fruit may reduce the content of stone cells [26]. Members of the *CAD* gene family have been identified in many species, such as the observation of nine *CAD* genes in *Arabidopsis* [16], 12 in rice [17], 15 in hybrid *Populus* (*Populus deltoides* × *Populus Nigra*) [14], and five in *Cucumis melo* [27]. However, few *AtCAD* genes are directly involved in lignification [10]. Indeed, only *AtCAD4* and *AtCAD5* are related to lignification [16]. In the present study, we combined the results pertaining to the physiological changes of stone cells with transcriptome data to identify nine genes that might be involved in lignin biosynthesis during pear fruit development. The nine candidate *PbrCAD* genes were confirmed by qPCR analysis, indicating that the expression levels of four genes were consistent with the changes in stone cell content. Therefore, we speculated that these four genes may participate in the regulation of lignin synthesis in pear fruit.

## 4. Materials and Methods

### 4.1. Collection and Identification of the CAD Genes

Chinese white pear (*Pyrus bretschneideri*) genome was retrieved from the Pear Genome Project (http://peargenome.njau.edu.cn/, 25 September 2019). Chinese plum (*P**runus m**ume*) sequence was obtained from the *Prunus mume* Genome Project (http://prunusmumegenome.bjfu.edu.cn/index.jsp, 25 September 2019). Apple (*Malus domestica*) and strawberry (*Fragaria vesca*) sequences were obtained from the Genome Database for Rosaceae (http://www.rosaceae.org/, 25 September 2019). Nine CAD protein sequences of *Arabidopsis* (AT1G72680, AT2G21730, AT2G21890, AT3G19450, AT4G34230, AT4G37970, AT4G37980, AT4G37990, and AT4G39330) were downloaded from The Arabidopsis Information Resource (http://www.arabidopsis.org/, 25 September 2019) and used as queries to perform a BLAST search against the four Rosaceae genome databases. Additionally, the CAD domains (PF08240 and PF00107) obtaining from the Pfam database (http://pfam.sanger.ac.uk/, 25 September 2019) were used to build an HMM file with HMMER3 (http://hmmer.org/, 25 September 2019). Subsequently, HMM searches were performed against the local protein databases of the four Rosaceae species using HMMER3. A total of 234 candidate *CAD* genes were identified from the four Rosaceae species. Then, the Pfam and SMART databases (http://smart.embl-heidelberg.de/, 25 September 2019) were used to check each candidate CAD-related protein sequence as a member of the *CAD* gene family. At last, the candidate genes which did not harbor the CAD domains or exhibit the domains with an incomplete structure were removed.

### 4.2. Phylogenetic Analysis

The CAD proteins of pear, Chinese plum, apple, and strawberry were aligned using ClustalX2.1 (http://clustalx.software.informer.com/2.1/, 25 September 2019). MEGA 7.0 was used to generate a phylogenetic tree of the *CAD* genes in Rosaceae [28]. The neighbor-joining (NJ) method was applied to construct various *CAD* trees. The bootstrap analysis was conducted with 1000 replicates.

### 4.3. Chromosomal Locations and Duplication Analysis

Chromosomal locations of the *CAD* genes were obtained on the basis of genome annotation data. The location data were plotted using Circos [29]. A method similar to PGDD (http://chibba.agtec.uga.edu/duplication/, 25 September 2019) was used to analyze the synteny [30,31]. BLASTP was used to search for potential homologous gene pairs (E < 1 × 10^−5^, top five matches) across multiple genomes. Finally, the homologous pairs were input in MCScanX to identify syntenic chains [32,33].

### 4.4. Calculation of Non-Synonymous (Ks) and Synonymous (Ka) Substitutions

To annotate the Ka and Ks substitution rates of syntenic gene pairs, the downstream analysis tools of MCScanX were used [34]. The Ka/Ks ratios were calculated using the toolbox KaKs_Calculator 2.0 and the Nei-Gojobori (NG) method [35,36].

### 4.5. Gene Structure and Motif Analyses in Pear

Structures of the *CAD* genes were obtained through alignment of ORFs and genomic sequences using the Gene Structure Display Server (http://gsds.chi.pku.edu.cn/, 25 September 2019) [37]. Sequences of PbrCAD proteins were elucidated using the online MEME website (http://meme-suit.org/tools/meme, 25 September 2019) [38]. The parameters were as follows: maximum number of motifs, 25; minimum motif width, 6; and maximum motif width, 200.

### 4.6. Plant Materials

This research used 40-year-old pear (*Pyrus bretschneideri* ‘Dangshansuli’) trees cultivating in a commercial orchard in Gaoyou, Jiangsu Province, China. A total of 10 strong and healthy trees were selected. On April 4th, 2016 (when the fruit trees were in the full-bloom stage), the branches with consistent bud growth and size in the middle of the canopy were selected and labeled. Fruit samples were collected every 8 days (starting at 15 days after flowering), and 30 fruits with a relatively consistent size were collected every time. All fruits were placed into ice box and transferred to the laboratory for further experimentation.

### 4.7. Measurement of Stone Cell Number

Stone cells were separated and counted using the method reported by Tao et al. (2009) and Cai et al. (2010) [5,20]. A 100-g sample of pear flesh was collected and stored at −20 °C for 24 h, followed by homogenization at 18,000 rpm for 5 min in distilled water. The suspension was stirred for 3 min, then precipitated at room temperature for 30 min. The supernatant was discarded, and the precipitate was suspended in 0.5 mol/L HCl for 30 min and washed with distilled water. This process was repeated several times until the stone cells separating from impurities. The assay was repeated three times.

### 4.8. Histochemical Staining

Freehand sections of fruit material were stained with 1% (*w*/*v*) phyloroglucinol dissolved in 95% ethanol. After 2 min, a drop of 35% HCl was added. The section was placed into a shadowless light box and photographed by a digital camera [5].

### 4.9. Expression Analysis Based on Transcriptomic Data

To further investigate the role of CAD family genes in the formation of pear stone cells, we analyzed the expression patterns of the PbrCAD genes based on transcriptome sequencing (RNA-Seq) data from previous studies [19,20,39]. The accession number of the transcriptomic data is PRJNA185970 on NCBI. We used the samples of 21, 35, 48, 55, 68, 85, and 114 days after full bloom (DAFB) to examine the expressions of *PbrCAD* genes. Total RNAs were extracted for RNA sequencing, and a sequencing library was constructed according to the manufacturer’s instructions (Illumina). The cDNA library was sequenced on an Illumina HiSeq™ 2000 sequencer (San Diego, CA, USA) without biological replicates.

### 4.10. Quantitative Real-Time PCR (qRT-PCR)

According to the manufacturer’s instructions, gene quantifications were performed using the SYBR Green Master Mix (SYBR Premix EX Taq, TaKaRa, Dalian, China). The mixture comprised 5.5 µL of nuclease-free water, 12.5 µL of 10× buffer, 0.5 M of each primer, and 1 µL of diluted cDNA. Each 20-µL reaction was run in triplicate. PCR was performed as follows: 5 min of incubation at 95 °C, followed by 45 cycles of 94 °C for 10 s, 60 °C for 30 s, and 72 °C for 30 s, and final extension at 72 °C for 3 min. *TUB-b2* (accession number, AB239681) was used to evaluate qRT-PCR. The relative mRNA levels were calculated using the relative 2^−^^ΔΔCT^ method [40]. The primers used to investigate the gene expression patterns in fruit tissues were listed in Table 4. Each RT-qPCR was performed in triplicate. Statistical analyses were performed using SPSS. In graphs, each bar represents the mean and standard error (SE) (*n* = 3).

## 5. Conclusions

In summary, a comprehensive analysis of the *CAD* gene family in Rosaceae was performed. A total of 149 full-length genes were identified and classified into six subgroups. A chromosome distribution analysis showed that the *CAD* genes were unevenly distributed in Rosaceae. Dispersed duplication and WGD/segmental duplication played the most important role in the expansion of the *CAD* gene family. The Ka/Ks ratio showed that five collinear *PbrCAD* gene pairs were under positive selection. *PbrCAD1*, *PbrCAD20*, *PbrCAD27*, and *PbrCAD31* were identified using qPCR may be involved in lignin synthesis in pear. Future research needs to identify the functions of these genes, and this study will be useful for elucidating functions of the *CAD* genes in pear.

## Figures and Tables

**Figure 1 plants-10-01444-f001:**
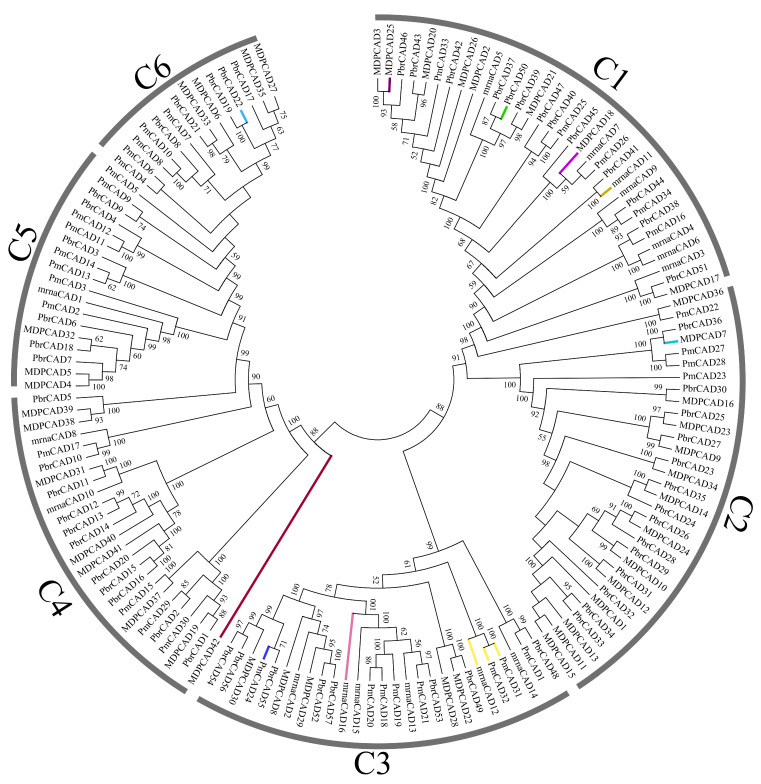
Neighbor-joining tree of CAD proteins in Rosaceae. The phylogenetic tree was constructed using the MEGA 7.0 based on the full-length amino-acid sequences encoded by CAD genes from *Pyrus bretschneideri* (57), *Malus domestica* (42), *Prunus mume* (34), and *Fragaria vesca* (16). A bootstrap analysis was performed with 1000 replicates.

**Figure 2 plants-10-01444-f002:**
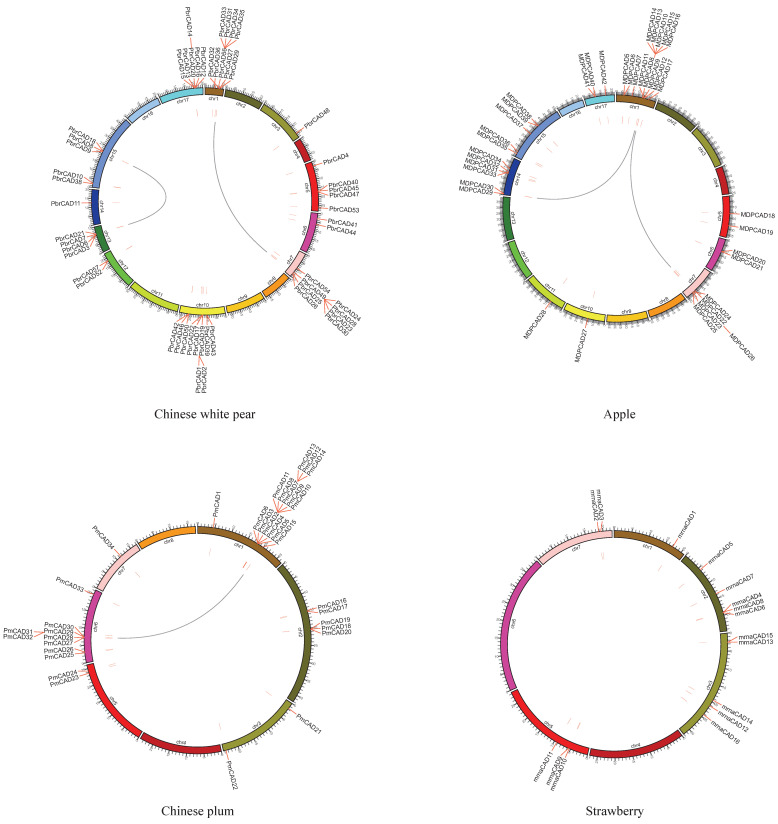
Chromosomal distribution and synteny of the *CAD* genes in Rosaceae genomes. The *CAD* genes in pear (*Pbr*), apple (*MDP*), Chinese plum (*Pm*), and strawberry (*mrna*) were mapped onto different chromosomes. Gene pairs with a syntenic relation-ship were joined by a line.

**Figure 3 plants-10-01444-f003:**
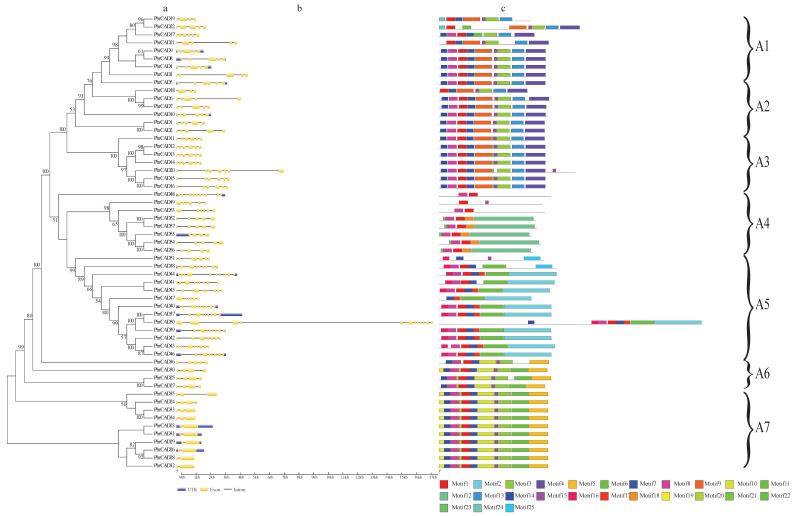
Phylogenetic tree, exon-intron structure, and conserved motifs of *PbrCAD* genes and proteins. (**a**) Unrooted neighbor-joining phylogeny of *PbrCAD*s, with bootstrap values >50. (**b**) The exon-intron structure is presented by yellow boxes corresponding to exons and the linking black lines corresponding to introns, while the blue line indicates the 5′-UTR and 3′-UTR. (**c**) Twenty-five conserved motifs identified by MEME tools in *PbrCAD* genes. Each colored box represents conservation (color figure online).

**Figure 4 plants-10-01444-f004:**
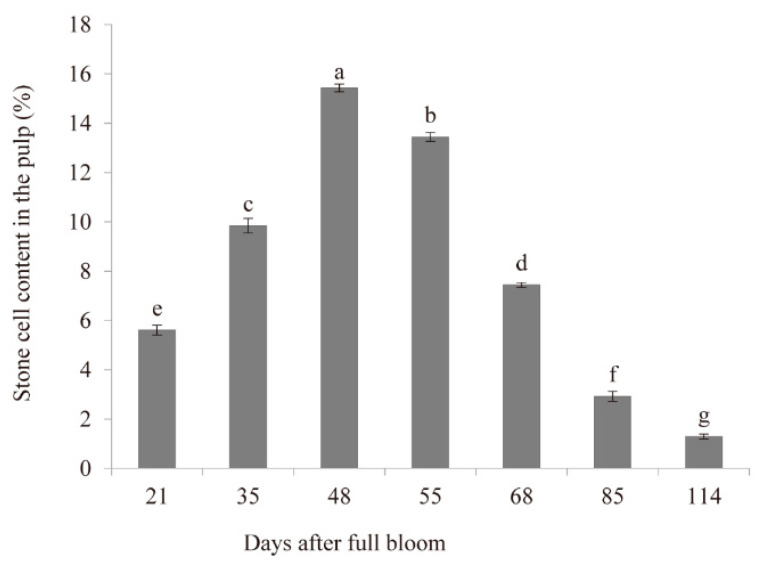
Changes in the stone cell content during fruit development. All data are reported as the average of three repeated exper-iments. Different letters indicate a difference of 5% in the significance levels. The vertical bars represent the SE.

**Figure 5 plants-10-01444-f005:**
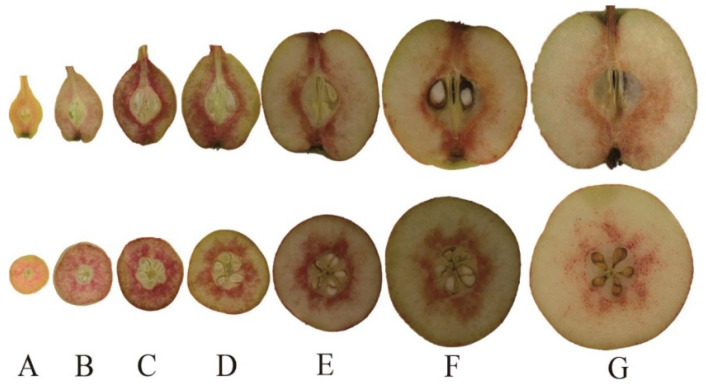
Histochemical staining with phloroglucinol-HCl. A-G correspond to the seven fruit developmental stages. (**A**) 21 days after full bloom (DAFB); (**B**) 35 DAFB; (**C**) 48 DAFB; (**D**) 55 DAFB; (**E**) 70 DAFB; (**F**) 85 DAFB; and (**G**) 114 DAFB. Red color indicates lignin deposition in pear fruit.

**Figure 6 plants-10-01444-f006:**
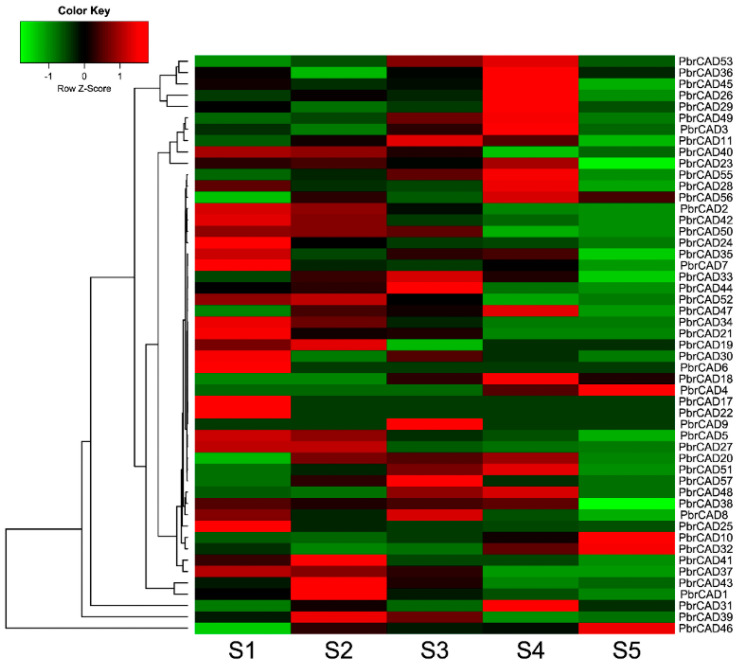
Heatmap of the expression levels of *PbrCAD* genes in Chinese white pear. Transcriptome sequencing was used to measure the expression levels of PbrCAD genes. S1-S5 correspond to the five fruit developmental stages: 15 days after full bloom (DAFB) (S1), 36 DAFB (S2), 81 DAFB (S3), 110 DAFB (S4), and 145 DAFB (S5). The color scale presented at the top of the figure represents the log2-transformed RPKM values. The light-green color indicates a low expression level and the red color indicates a high ex-pression level.

**Figure 7 plants-10-01444-f007:**
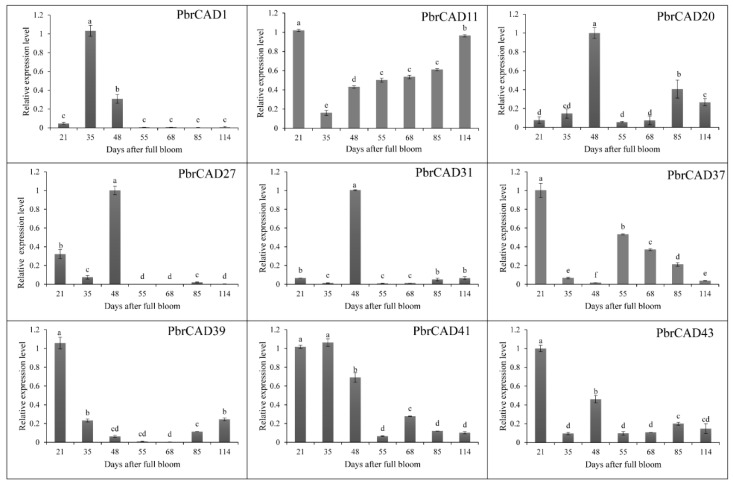
Quantitative real-time PCR was used to detect the expression of nine *PbrCAD* genes during fruit development. All data are reported as the average of three repeated experiments. The different letters indicate a difference of 5% in the level of sig-nificance. The vertical bars represent the SE.

**Table 1 plants-10-01444-t001:** Genome information and number of *CAD* genes in Rosaceae species.

Common Name	Species Name	ChromoSome Number	Release Version	Genome Gene Number	Identified *CAD* Genes	Gene Name Prefix
**Pear**	*Pyrus bretschneideri*	34	NJAU, v1.0	42341	57	Pbr
**Apple**	*Malus domestica*	34	GDR, v1.0	63541	42	MDP
**Chinese plum**	*Prunus mume*	16	BFU, v1.0	31390	34	Pm
**Strawberry**	*Fragaria vesca*	14	GDR, v1.0	32831	16	mrna

**Table 2 plants-10-01444-t002:** The numbers of *CAD* genes of different origins in four Rosaceae species.

Scheme	Number of *CAD* Genes	Number of CAD Genes of Different Origins (Percentage)				
		Singleton	WGD	Tandem	Proximal	Dispersed
**Pear**	57	8 (14.0)	24 (42.1)	0	0	25 (43.9)
**Apple**	42	0	2 (4.8)	0	0	40 (95.2)
**Chinese Plum**	34	0	13 (38.2)	0	0	20 (58.8)
**Strawberry**	16	0	9 (56.2)	0	0	7 (43.8)

**Table 3 plants-10-01444-t003:** Estimated divergence period of the *PbrCAD* gene pairs. Ks, synonymous substitution rate; Ka, non-synonymous substitution rate; MYA, million years ago; NG, Nei-Gojobori.

Common Name	Method	Ka	Ks	Ka/Ks	*p* Value (Fisher’s Test)
*PbrCAD19*- *PbrCAD22*	NG	0.030518	0.0331618	0.920278	0.822196
*PbrCAD24*- *PbrCAD35*	NG	0.03653	0.294749	0.123935	1.02 × 10^−21^
*PbrCAD25*- *PbrCAD27*	NG	0.014796	0.186026	0.0795362	8.38 × 10^−18^
*PbrCAD26*- *PbrCAD29*	NG	0.019842	0.249629	0.0794857	8.57 × 10^−24^
*PbrCAD39*- *PbrCAD50*	NG	0.036999	0.235759	0.156936	1.28 × 10^−16^

**Table 4 plants-10-01444-t004:** List of primers used for quantitative real-time PCR analyses of *PbrCAD* and internal control genes.

Gene Name	Forward Primer Sequence (5′ → 3′)	Reverse Primer Sequence (5′ → 3′)	Amplicon Length (bp)
*PbrCAD1*	TGACCTTGGCACGTCAAACT	CAGTACTGCTCGTTGTCCGT	174
*PbrCAD* *11*	CGGAACAAAGGACACGCAAG	TCGAGCGCTTCAGTTGCATA	102
*PbrCAD20*	CCAGGCCGGAAATTCACTG	TGCCGTAAAGAGTTGTATCAGC	221
*PbrCAD27*	GGGCCCATGATGTTCGAGT	AACTTCATGTCCGGGCAAAGA	245
*PbrCAD31*	TGTTAGAGACGCCAAACCTGC	TCCGATCACCATCGGATCCTTA	195
*PbrCAD3* *7*	GTCTACAGCTGGTCAGGTTATCAGATG	CCACAACTCCTCCAGCTTCATGA	202
*PbrCAD39*	GATGGCCAGTCCAGGTTCTC	AGCAAGGGAACCAACATGAC	102
*PbrCAD41*	TCAGCTCACTTGTGCCTCTG	CACCTTGTGCAACGGAAAGG	153
*PbrCAD43*	GTCCAAGTCGAGGTGGCACC	TGCATTCTCCTGTGAACACTGGCA	203
*TUB-b2*	TGGGCTTTGCTCCTCTTAC	CCTTCGTGCTCATCTTACC	169

## Data Availability

All data, tables, figures, and results in paper are our own and original.

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
