# Peer review of "CAD Genes: Genome-Wide Identification, Evolution, and Their Contribution to Lignin Biosynthesis in Pear (Pyrus bretschneideri)"

_plants, 2021, doi:10.3390/plants10071444_

Round 1
Reviewer 1 Report
The authors did some revisions, and the study improved. The professional content is acceptable, but still there are points that need to be revised. I think the study is suitable for publication after some further revisions.
1) The letter sizes are not uniform in the text (e.g. L 48-51, L102, L194-195, L215-216, L248, L313-314).
2) Figures and tables should be inserted in the text. In this version figures are not included (I was not able to see in the downloaded pdf file). I saw figures in the previous version of this manuscript. They were in general OK but I had some suggestion for them in my previous review, but now I can not see whether the authors did changes or not (as I can not see the figures in the present manuscript pdf file).
3) L408: Pyrus bretschneideri – in italic
Author Response
1) The letter sizes are not uniform in the text (e.g. L 48-51, L102, L194-195, L215-216, L248, L313-314).
Answer:
This suggestion is very good, we have changed them.
2) Figures and tables should be inserted in the text. In this version figures are not included (I was not able to see in the downloaded pdf file). I saw figures in the previous version of this manuscript. They were in general OK but I had some suggestion for them in my previous review, but now I can not see whether the authors did changes or not (as I can not see the figures in the present manuscript pdf file).
Answer:
This suggestion is very good. We have reorganized the article following the instruction of ‘Plants’, and have put the figures and tables in ms.
3) L408: Pyrus bretschneideri – in italic
Answer:
Thank you for reminding and we have revised it.

Reviewer 2 Report
The author had revised the manuscript accordingly. I very much appreciate of it. However, several minor points should be addressed before publication.
1) A proof reading is highly required.
2) Please format the manuscript as 'Plants' formats (Second time I mentioned).
3) L82-83, L1176-178, L193-195, L216, L322-325. The font size and format is different.
4) L137-138, L147-149. Delete. These should be mentioned and repeated with Ms&Ms
Author Response
Response to reviewer 2:
1) A proof reading is highly required.
Answer:
Thank you for your suggestion. We have checked the English in the ms, and invited an English professional to revise language, and the changes were highlighted in yellow.
2) Please format the manuscript as 'Plants' formats (Second time I mentioned).
Answer:
This suggestion is very good. We have reorganized the article following the instruction of ‘Plants’, and have put the figures and tables in ms.
3) L82-83, L1176-178, L193-195, L216, L322-325. The font size and format is different.
Answer:
This suggestion is very good, we have changed them.
4) L137-138, L147-149. Delete. These should be mentioned and repeated with Ms&Ms
Answer:
Thank you. We have deleted them.
